# Immunohistochemical Study of Bladder Cancer Molecular Subtypes and Their Association with PD-L1 Expression

**DOI:** 10.3390/cancers15010188

**Published:** 2022-12-28

**Authors:** Dimitrios Goutas, Kostas Palamaris, Anastasios Stofas, Nektarios Politakis, Antonia Despotidi, Ioanna Giannopoulou, Nikolaos Goutas, Dimitrios Vlachodimitropoulos, Nikolaos Kavantzas, Andreas C. Lazaris, Hariklia Gakiopoulou

**Affiliations:** 1First Department of Pathology, School of Medicine, The National and Kapodistrian University of Athens, 115 27 Athens, Greece; 2Department of Forensic Medicine and Toxicology, School of Medicine, The National and Kapodistrian University of Athens, 115 27 Athens, Greece

**Keywords:** bladder cancer, molecular subtypes, immunotherapy, basal, luminal, PD-L1

## Abstract

**Simple Summary:**

The aim of our study was to stratify bladder cancer patients into their molecular subtypes using a simple and inexpensive immunohistochemical algorithm and further provide any associations with PD-L1 expression. Given the fact that there is a universal lack of predictive biomarkers for immunotherapy, we suggest the possibility of stratifying patients into likely-responders and likely-not-responders to anti-PD-L1 therapy, based on their bladder cancer molecular subtypes.

**Abstract:**

The significant heterogeneity in clinical outcomes among patients with bladder cancer has highlighted the existence of different biological subtypes of muscle-invasive bladder cancer (MIBC) and non-muscle-invasive bladder cancer (NMIBC). Meanwhile, immune checkpoint proteins and their interference with tumor-related immune-evasive strategies has led to the development of several immunotherapeutic drugs targeting programmed death-1 (PD-1) or programmed death ligand-1 (PD-L1). However, the lack of any known biomarker that could predict responses to immunotherapy has led to a more agnostic therapeutic approach. Here, we present a study conducted in 77 bladder cancer (BC) patients (n = 77), ranging from stages pTa to pT2. Tumor specimens were resected via transurethral resection of bladder tumor (TURBT) and consistuted of 24 low-grade (LG) and 53 high-grade (HG) tumors. Patients’ tumors were then categorized into molecular subtypes, via immunohistochemistry (CK5/6 and GATA3). Furthermore, all tumor specimens were stained with anti-PD-L1 and demonstrated significant correlations with basal immunophenotype, stage pT2 and HG tumors. As such, we attempted to stratify patients into groups of likely-responders and likely-not-responders to immunotherapy with anti-PD-L1, based on their molecular phenotype. Finally, in acknowledging the fact that there is a universal lack of biomarkers associated with predicting BC response to immunotherapeutic drugs, we tested all tumors for deficiency of mismatch repair proteins (MMR).

## 1. Introduction

Bladder cancer (BC) is the 12th most common malignancy and the most frequent malignant tumor of the urinary system, with over 500.000 cases diagnosed every year [1]. While primary bladder carcinomas are characterized by some histological heterogeneity, the majority of them display urothelial differentiation [1]. Based on their depth of invasion within bladder wall, they are divided into either non-muscle-invasive bladder carcinoma (NMIBC) or muscle-invasive bladder carcinoma (MIBC). NMIBC accounts for approximately over 70% of all newly diagnosed BCs [1], including non-invasive papillary urothelial carcinomas (stage pTa), carcinoma in situ (CIS) and invasive carcinomas limited to the lamina propria (stage pT1) [1]. Treatment approaches for NMIBCs usually include transurethral resection of the visible tumor, followed by intravesical Bacillus Calmette-Guerin (BCG) therapy when the tumor contains high-risk features, such as high-grade histology or lamina propria invasion [1]. MIBCs include carcinomas that invade the detrusor muscle and beyond, with therapy consisting of cystectomy preceded by neoadjuvant chemotherapy (NAC) or chemoradiation [1]. Following the paradigm set by other tumors, immunotherapy with blockade of immune checkpoint inhibitors (CPIs) has also begun to be adopted, predominantly as a second-line treatment option in cases that are unresponsive to conventional BCG/chemotherapeutic schemes, as well as in patients unwilling or unable to undergo surgical resection. In fact, currently, there are five PD-1/PD-L1 inhibitors that are approved for treatment of locally advanced or metastatic urothelial carcinoma of the bladder and the upper urinary tract [1].

Recent development of molecular biology, which enables more in-depth large-scale analyses, has led to the further subdivision of specific histological tumor categories into distinct molecular subtypes, which constitute distinct entities with unique gene mutations, copy-number aberrations, DNA methylation and RNA expression patterns. In this setting, transcriptional profiling of human primary bladder cancer specimens have been classified into ‘intrinsic’ basal and luminal molecular subtypes [1]. Luminal tumors have a papillary configuration and express markers of urothelial differentiation (uroplakins, cytokeratin 20), GATA3, fibroblast growth factor 3 (FGFR3), E-cadherin and early cell-cycle genes [1]. On the contrary, basal tumors express markers of the basal layer of the urothelium (cluster of differentiation 44 (CD44), cytokeratin 5/6 (CK5/6) and cytokeratin 14 (CK14)), with some exhibiting squamous differentiation [1]. Further studies aiming to accurately classify urothelial carcinomas into molecular subtypes have been performed by various research groups [1,2,3,4,5,6,7]. All the subtypes recognized by the groups revealed substantial concordance among them. In 2019, an effort to achieve an international consensus on MIBC was made using 1750 MIBCs from 16 published datasets and two additional cohorts to compare six molecular classification schemes [8]. Kamoun et al. [8] defined MIBCs with six molecular subtypes, labeled as luminal papillary (LumP), luminal nonspecified (LumNS), luminal unstable (LumU), stroma-rich, basal/squamous (Ba/Sq) and neuroendocrine-like (NE-like). They further analyzed seven bladder cancer genes that were most mutated (CDKN2A, FGFR3, proliferator-activated receptor gamma (PPARG), human epidermal growth factor receptor 2 (HER2; ERBB2), TP53, E2F3 and RB1), and generated comprehensive profiles of their genomic alterations for each of the above subtypes [8]. The distinct molecular subtypes of bladder cancer are associated with perceived differences in their responses to targeted and non-targeted therapies and statistically significant variations in their clinical outcomes. The worst prognosis is observed in tumors of NE-like and Ba/Sq subtypes, while among the remaining four molecular classes, LumU is characterized by a more dismal prognosis. LumP, LumNS and stroma-rich have similar survival rates. Regarding response to systemic treatment, no correlation is encountered between NAC or immune checkpoint immunotherapy response and molecular subtypes.

Considering the fact that current immunotherapeutic agents are only effective in a restricted fraction of patients, the identification of specific parameters that can serve as potential predictors of response to CPIs is urgently needed. Cumulative knowledge from heterologous malignancies have identified PD-L1 expression levels and mismatch repair system deficiency (MMR-D) as pivotal biomarkers, capable of indicating the level of susceptibility to CPIs [9,10,11,12]. At the moment, four PD-1/PD-L1 detection assays are employed in clinical trials: 28-8 PD-1 pharmDx (Nivolumab), 22C3 PD-1 pharmDx (Pembrolizumab), Ventana PD-L1 SP142 (Atezolizumab), and Ventana PD-L1 SP263 assays (Durvalumab). While PD-L1 expression is considered a pre-requisite for the effective utilization of CPIs, it cannot be considered a reliable criterion for the selection of patients who will benefit from CPIs. The restrictions of PD-L1 utility as a predictive biomarker can be attributed to a number of factors, including the inconsistency in the definition of PD-L1 positivity, and the variations encountered among different standard assays for PD-L1 expression, and different antibody clones, which result in notable heterogeneity of data. Moreover, the deployment of PD-L1 expression as a selection tool in routine clinical practice should consider the temporal heterogeneity, which can be observed among different tumor stages. For example, substantial discordance has been observed between PD-L1 expression in lymph node metastatic foci and corresponding cystectomy samples. As a result, PD-L1 evaluation in prospective candidates for immunotherapy should be conducted in samples collected immediately before treatment initiation rather than in archived tissue specimens. The mismatch repair (MMR) system is vital for the rectification of DNA sequence mismatches during DNA replication, and, as a result, loss of function of one of the MMR proteins (*MLH1, MSH2, MSH6, PMS2*) leads to high rates of mutations that accumulate in repetitive nucleotide regions (microsatellites) [13]. Microsatellite instability (MSI), also termed MMR deficiency (MMR-d), could often display oncogenic potential, when it comes to coding regions of genes involved in critical cellular functions [14]. Epigenetic inactivation of *MLH1* and *MSH2* represent the majority of sporadic MSI tumors [15,16]. Furthermore, MMR-d tumors have 10-100 times more somatic mutations than microsatellite-stable (MSS) tumors, leading to an increased neoantigen burden and immunogenicity [17]. As a result, MMR-d tumors are known to be responsive to the anti-programmed death (PD)-1/programmed death ligand (PD-L)-1 antibodies [18]. In bladder carcinomas, MMR-d is only encountered in a minor proportion of cases, often developed in a Lynch Syndrome background [19], and it seems to be an early event in the multi-step process of tumorigenesis, as specimens display diffuse homogenous MMR protein defects, mainly PMS2, MLH1 and MSH2 [16].

The present study aimed to stratify BC patients into their molecular subtypes and observe whether certain molecular phenotypes would display a stronger association with PD-L1 expression in such way that these patients could be promoted for anti-PD-L1 therapy. Furthermore, we tested the immunohistochemical expression of MMR proteins (*MLH1, MSH2, PMS2, MSH6*) and tried to establish their role as biomarkers for immunotherapy response in BC (Figure 1).

## 2. Materials and Methods

### 2.1. Patient Material Collection and Characterization

Formalin-fixed paraffin-embedded (FFPE) tissues from 77 transurethral resections of bladder tumors (TURBT) of stages pTa, pT1 and pT2 were retrieved from the pathology laboratory archives of Laikon General Hospital in Athens, Greece. Females represented the 20.8% of our samples and males the 79.2%. Tumor stages were 23.4% pTa, 36.4% pT1 and 40.3% pT2, while 31.2% were LG tumors and 68.8% were HG tumors.

### 2.2. Immunohistochemistry Procedure and Evaluation

Immunohistochemistry was carried out by using standard procedures in the 77 tissue specimens. Firstly, the sections were stained with the following antibodies on a Dako system, according to the manufacturer’s protocol: CK5/6 (clone D5/16 B4, Dako/dilution 1:150, Carpinteria, CA, USA), and then GATA3 (clone L50-823, Dako/dilution 1:500, Carpinteria, CA, USA). After a literature review, we used a simple algorithm for subtyping NMIBC and MIBC into their molecular subtypes, based on their morphology and two common antibodies: GATA3 and CK5/6 [1,7,8,20,21,22,23]. The positivity cut-off value was 20% with at least moderate intensity, as recommended by Dadhania et al. [24], and samples were considered negative if the immunohistochemical expression of the target proteins was less or equal to 20% and positive if their expression was more than 20%.

Secondly, the sections were stained with antibodies against MLH1 (clone ab92312, AbCam/at dilution 1:200, UK), PMS2 (clone ab110638, AbCam/at dilution 1:200, UK), MSH6 (clone ab92471, AbCam/at dilution 1:200, UK) and MSH2 (clone 227941, AbCam/at dilution 1:200, UK). Antigen retrieval was performed at pH 6. The Envision (Dako) visualization system was used. DAB (3,3-diaminobenzidine) was used as a chromogen and hematoxylin as counterstain.

Finally, the sections were stained with a rabbit monoclonal antibody against PD-L1 (clone ZR1, Zeta dilution 1:100, Southern California). PD-L1 immunohistochemical interpretation was made by using the three established available methods [19,25]: by calculating the percentage of immune cells’ positive membranous staining, per tumor area (IC score); by calculating the percentage of tumor cells’ positive membranous staining per the number of viable tumor cells (tumor proportion score, TPS); and by calculating the percentage of positive membranous staining in both immune cells and tumor cells, divided by the number of viable tumor cells (combined positive score, CPS). Using the IC score, we evaluated the immune cells present in the intratumoral and contiguous peritumoral stroma, with the stain highlighting a heterogeneous population of cells, the majority of which being lymphocytes, macrophages, dendritic cells, and granulocytes [19,25]. The immunohistochemical interpretation method is summarized in Figure 2. As most studies that led to FDA approval of anti-PD-L1 agents in BC used a cutoff value of 1%, we also used this value for statistical analysis purposes [26,27,28].

### 2.3. Statistical Analysis

Initial statistical analysis was based on descriptive statistics. Categorical variables were described with absolute and relative frequencies, while, for continuous variables, mean, standard deviation, median, minimum, and maximum were provided. The association of subtype with sex, stage and grade was based on Pearson chi-square test without continuity correction. PD-L1 Immune Cells (%), PD-L1 TPS and PD-L1 CPS, were categorized as positive and negative expression, alongside considering the 1% cut-off based on previous published data [26,27,28]. The chi-square test without continuity correction was also used for the association of positive and negative expressions with subtype, sex, grade, and stage. To avoid potential confounders, multivariate logistic regression was employed for PD-L1 Immune Cells (%), PD-L1 TPS and PD-L1 CPS. Statistical analysis was carried out in SPSS Version 20 and the statistical significance was set to α = 0.05.

### 2.4. Database and TCGA Analysis

We used cBioPortal to identify genetic alterations (deletion) to the PMS2 gene and to detect CD274 (PD-L1) overexpression in the bladder carcinomas cohort of The Cancer Genome Atlas (TCGA). We also utilized Gene Expression Profiling Interactive Analysis (GEPIA), a database that retrieves data from TCGA tumor samples, in order to identify potential correlations of PD-L1 expression levels with basal and luminal gene signatures. Moreover, the survival analysis module of the GEPIA2 database was used to analyze the association between the expression of PMS2 and patients’ overall and disease-free survival (OS and DFS). Kaplan-Meier curves were generated by employing a median cutoff value of 50% and the log-rank *p*-value was calculated.

## 3. Results

### 3.1. Molecular Taxonomy

Patients who were CK5/6 positive and GATA3 negative were classified as basal, constituting 29,9% of our cohort (Table 1) (Figure 3), and patients who were CK5/6 negative and GATA3 positive were classified as luminal, constituting 70.1% of our cohort (Table 1) (Figure 4) [24]. As far as statistical correlations of basal and luminal tumors with clinicopathologic parameters are concerned, statistically significant associations emerged among basal phenotype and histological stage (*p* < 0.001) and tumor grade as well (*p* = 0.025). Notably, patients with basal tumors had an almost 17 times higher chance of having pT2 bladder tumors and a 4.24 times higher risk of high-grade bladder carcinomas when compared with patients with luminal tumors (Table 2 and Table 3).

### 3.2. Mismatch Repair Deficiency (MMR-d)

Immunohistochemical interpretation of MMR protein expression in our cases revealed a strong and uniform expression in all tissue samples but one (n = 77). More specifically, one case demonstrated heterogeneous PMS2 expression loss (Figure 5) while MLH1 had retained its expression. The low frequency of PMS2 loss was also confirmed in the cohort of bladder urothelial carcinomas of The Cancer Genome Atlas (TCGA), as deletion occurred in just 2/408 cases. In both specimens, deletion was homozygous. Regarding PMS2 association with prognosis, reduced PMS2 expression levels were associated with longer overall survival (OS) and disease-free survival (DFS) (Figure 6), even though the correlation was not statistically significant (*p* - values of 0.071 and 0.06 respectively).

### 3.3. PD-L1 Assessment

Interestingly, we observed that PD-L1 positive expression showed a strong and consistent association with patients who had basal type tumors (Figure 7 and Figure 8). Furthermore, another interesting correlation was the percentage of positive PD-L1 expression (by using the IC score algorithm) in patients with tumor stages pTa/pT1 (Figure 9). Finally, assessment of all scoring algorithms provided significant associations with tumor stage (pT2) and grade (HG) (Figure 10). In multivariate logistic regression analysis, grade and stage variables remained statistically significant in their association with TPS PD-L1 expression, with odds ratios 3.83 and 4.95, respectively, and *p*-values 0.049 and 0.024, respectively. The higher expression levels of PD-L1 in tumors of basal signature, as compared with luminal ones, was confirmed in TCGA cohort. The correlation of PD-L1 with basal and luminal tumors was based on four-gene signatures consisting of well-established markers defining the two main molecular subtypes: luminal tumors were defined by GATA3, uroplakin 1A (UPK1A), uroplakin 3 (UPK3) and cytokeratin 20 (KRT20), while basal ones by cytokeratin 5, cytokeratin 6B, cytokeratin 14 and CD44 [21]. Indeed, basal gene signature was associated with increased PD-L1 expression (R = 0.26, *p*-value < 0.01), while an inverse correlation was observed between PD-L1 and the luminal signature (R = −0.36, *p*-value < 0.01) (Figure 11).

## 4. Discussion

Bladder cancer represents a highly heterogeneous tumor with inconsistent and abnormal protein expression of terminal differentiation markers, suggesting pseudo-differentiation [7]. As such, in global mRNA analyses, we can observe co-clustering of tumors with different cancer cell phenotypes and clustering-apart of tumors with identical cancer cell phenotypes [7]. This divergence/convergence advocates that global commonality may exist between MIBC tumors, regardless of specific tumor-cell phenotype. Therefore, subtype classification using global mRNA profiling and immunohistochemical profiling at the tumor-cell level results in a systematic disagreement. Sjodahl et al. [7] suggested the use of global mRNA profiling combined with molecular pathology for an adequate subtype classification of MIBC.

However, implementing this approach in an everyday pathologist’s routine is not realistic, while also carrying a high cost. This is especially if one considers the extensive overlapping that the molecular subtypes of these tumors display. As suggested by Dadhania et al. [24] patients who are CK5/6 positive and GATA3 negative can be classified as basal and patients who are CK5/6 negative and GATA3 positive can be classified as luminal. Several studies have reported the possibility of further subclassification into their molecular subtypes, as established in the recent consensus on molecular classification by Kamoun et al. [8], based on p16 and FGFR3 expression, as well as on their histomorphology [21,22]. More specifically, LumP are p16(−) and FGFR3(+) with a characteristic papillary configuration, while LumNS are p16(+) and FGFR3(−) and display a solid muscle-invasive histology. Finally, LumU tumors share the same immunophenotype with the LumP category, but they are linked with micropapillary urinary bladder carcinomas [8]. In our study, we did not try to distinguish immunohistochemically the luminal tumors into their subcategories, as the wide and complex molecular background in combination with the extensive molecular heterogeneity of bladder tumors^7^ would render our efforts inconsistent.

As such, we propose the use of a basic immunohistochemical panel [20,23] used by pathologists on a day-to-day basis, every time they encounter BC specimens. The use of CK5/6 and GATA3 to broadly classify patients into two umbrella categories of basal and luminal tumors, along with the evaluation of their histomorphology, can accurately enough depict their subtypes. Furthermore, adding the use of p16 and FGFR3 can provide a more specific molecular subclassification [8,20].

For the last few years, the BC landscape has remained relatively unchanged regarding the use of biomarkers that could predict immunotherapy response. Our study confirmed that MMR-d bladder tumors are present in a very small percentage and, as such, no correlations could be made with PD-L1 expression. Nevertheless, contrary to what Fraune et al. [29] reported in their study, regarding the heterogeneous expression of MMR proteins, and stating that any heterogeneous MMR expression is due to either biological reasons (fixation etc.) or due to errors in immunohistochemistry and various other artifacts, we suggest that heterogeneous expression loss of any of the MMR proteins does not necessarily link to an artifact or any other type of biological factors interfering with their expression, but with an MSI-high status in these areas. In these cases, further investigation with polymerase chain reaction (PCR) or next-generation sequencing (NGS) is warranted.

Some studies have suggested the use of tumor mutational burden (TMB) as a potential biomarker for immunotherapy, with, however, conflicting results, mostly regarding the challenges in unifying and standardizing the definition of mutation burden [30]. It goes without saying that standardized, reproducible biomarkers are needed to guide treatment decisions, as until now, not a single test has provided reproducibility to predict responders to immunotherapy.

In our study, assessment of all PD-L1 scoring algorithms (ICs, TPS, CPS) provided significant associations with tumor stage (pT2) and grade (HG), in agreement with previous studies [31,32,33]. Furthermore, significant association was observed among PD-L1 positivity and pTa/pT1 bladder cancer patients, with more than half of the patients showing a PD-L1 IC score >1%. However, although the correlation did not reach statistical significance, the high percentage of PD-L1-positive pTa/pT1 patients is suggestive that superficial BC patients could also benefit from immunotherapy. Currently, there is only one clinical trial running for pTa patients who are BCG ineligible [34].

Different prognoses and chemotherapy sensitivities have been reported to be associated with different molecular subtypes. Phase III CheckMate clinical trial reported different immunotherapy responses to Nivolumab, corresponding to separate molecular subtypes in patients with metastatic urothelial carcinomas, with the highest response being reported in the TCGA cluster III, which corresponds to the basal type [28]. Moreover, Hodgson et al. [35] demonstrated that MIBC could be classified into basal and luminal subtypes based on CK5/6 and GATA3 expression and each subtype was analyzed for its PD-L1 (SP263 clone) expression [35]. The results showed that basal type tumors revealed higher PD-L1 positivity, when compared with the luminal category [35]. The aforementioned findings are in line with our results, showing that basal type tumors demonstrate higher PD-L1 expression.

In summary, the evidence provided by our study, regarding the expression pattern of PD-L1 in luminal and basal bladder urothelial carcinomas, largely reconcile with the findings of previous analyses. However certain limitations derive mainly from the relatively small sample size and with the fact that our evaluation was based on purely immunohistochemical detection of the proteins of interest. Especially in the case of PD-L1, as described above in the “Introduction” section, significant deviations are observed between different cohorts, which can be attributed to thse different diagnostic assays and antibody clones, as well as in the divergent cut-off scores employed. Moreover, the lack of molecular analysis of our samples means that their categorization into luminal and basal groups was based on immunohistochemical detection of two markers. As a result, we were not able to incorporate the more precise, six-subtype molecular classification of BC. The data from immunohistochemical studies should be complemented by genomic/transcriptomic analyses, which can provide more in-depth insight into the correlations of specific genetic alterations and gene signatures with the PD-L1 expression profile. This could have great impact, from both a basic research and clinical perspective: it could help clarify the intracellular networks controlling PD-L1 levels and precisely identify additional biomarkers, with direct impact on its expression, thus creating predictive signatures of the immunotherapy response.

## 5. Conclusions

After elaborating on the above results and in conjunction with our PD-L1 clinicopathologic associations, we suggest the global use of a single and basic immunohistochemical panel (CK5/6, GATA3) for molecularly subtyping BCs and providing the possibility to stratify patients into groups of likely-responders and likely-not-responders to PD-L1, based on their molecular phenotype. Our study revealed a strong and consistent association of PD-L1 expression with patients of a basal immunophenotype, raising the possibility of these patients to respond well in therapy with anti-PD-L1. Further studies are to be conducted, containing larger population cohorts, in order to confirm the association among BC molecular subtypes and PD-L1 response.

## Figures and Tables

**Figure 1 cancers-15-00188-f001:**
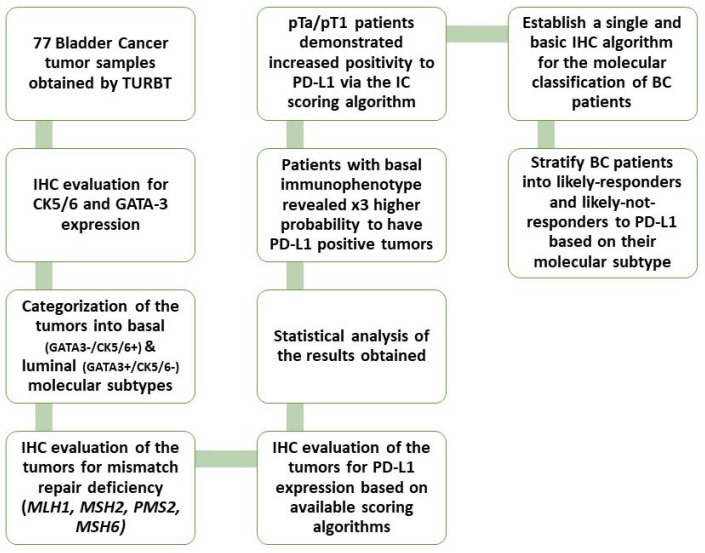
Workflow diagram of our study.

**Figure 2 cancers-15-00188-f002:**
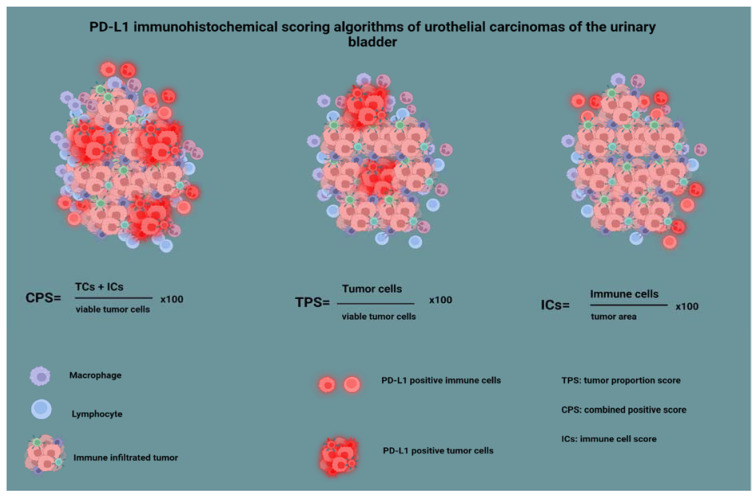
PD-L1 immunohistochemical scoring algorithms of urothelial carcinomas of the urinary bladder.

**Figure 3 cancers-15-00188-f003:**
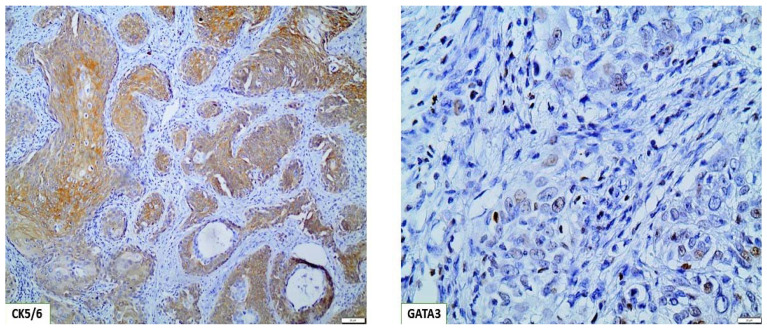
Basal/squamous BC molecular subtype, displaying strong CK5/6 expression and weak focal GATA3 expression (×100).

**Figure 4 cancers-15-00188-f004:**
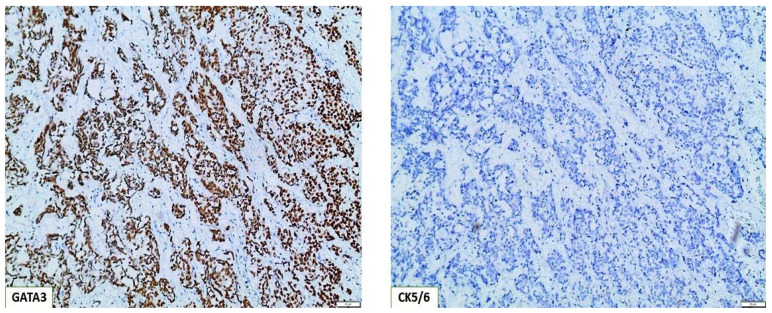
Molecular classification of basal type bladder tumors, based on the immunohistochemical algorithm GATA3(+)/CK5/6(−) (×100).

**Figure 5 cancers-15-00188-f005:**
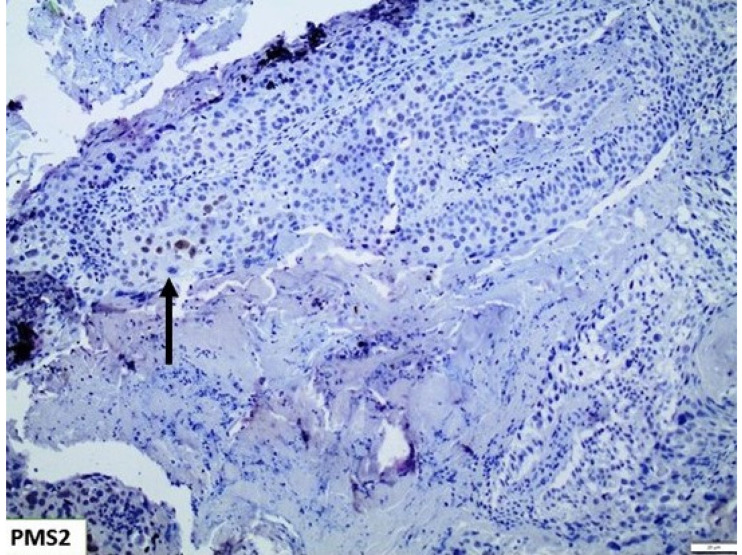
*PMS2* showing extensive loss of expression, with only few nuclei retaining its expression (*Arrow)* (×100).

**Figure 6 cancers-15-00188-f006:**
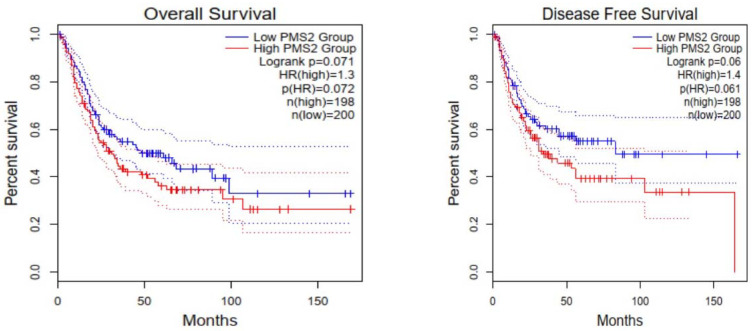
Cases with low PMS2 expression levels are characterized by longer overall survival (OS) and disease-free survival (DFS); however, this correlation is not statistically significant (*p*-values 0.071 and 0.06, respectively).

**Figure 7 cancers-15-00188-f007:**
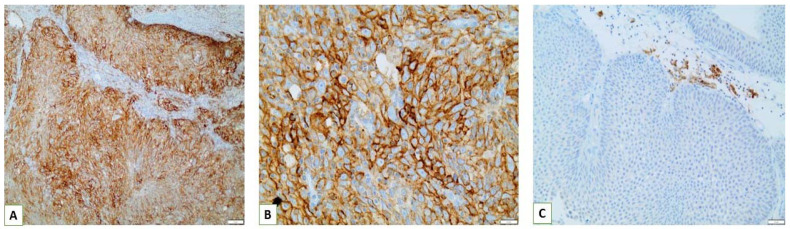
(**A**,**B**) Strong positive PD-L1 membranous expression in tumor cells of a basal type MIBC (×200, ×400); (**C**) Strong positive membranous expression of PD-L1 in immune cells present in the contiguous peritumoral stroma of pTa NMIBC (×200).

**Figure 8 cancers-15-00188-f008:**
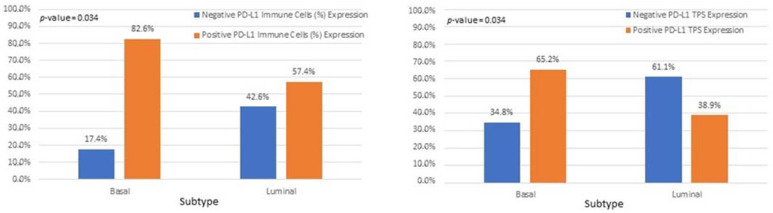
Associations of IC and TPS PD-L1 scores with bladder cancer molecular subtypes.

**Figure 9 cancers-15-00188-f009:**
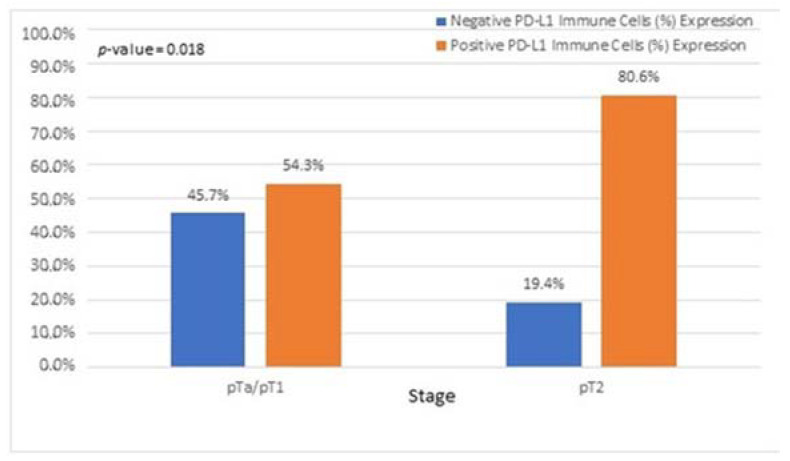
Associations of patients with pTa/pT1 tumors with PD-L1 IC score.

**Figure 10 cancers-15-00188-f010:**
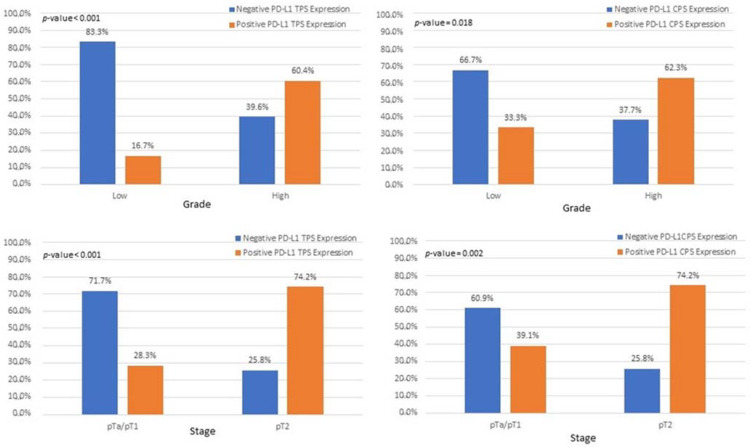
Associations among patients with pT2 and high-grade tumors with PD-L1 expression, based on CPS and TPS scores.

**Figure 11 cancers-15-00188-f011:**
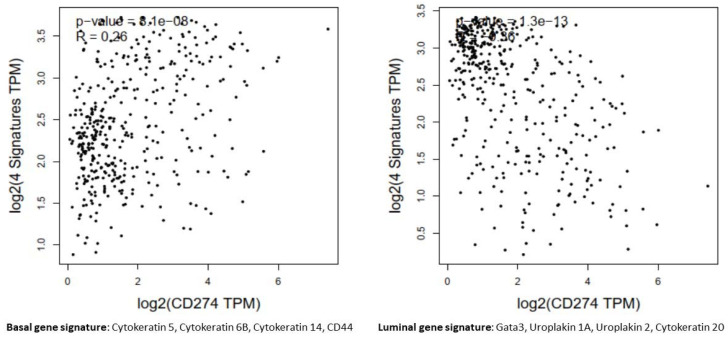
In the bladder carcinomas cohort of TCGA, PD-L1 (CD274) gene expression levels were positively correlated with markers of basal tumors (R = 0.26, *p*-value < 0.01) and inversely correlated with the luminal genes signature (R = −0.36, *p*-value < 0.01).

**Table 1 cancers-15-00188-t001:** Clinicopathological features of obtained bladder cancer samples.

Study Variable	Frequency (N = 77)	Percent (%)
Subtype		
Basal	23	29.9
Luminal	54	70.1
Sex		
Female	16	20.8
Male	61	79.2
Grade (WHO 2022)		
Low	24	31.2
High	53	68.8
Stage		
pTa	18	23.4
pT1	28	36.4
pT2	31	40.3

**Table 2 cancers-15-00188-t002:** Association of BC molecular subtypes with stage.

*p*-Value < 0.001 OR = 16.63, 95% CI: (4.74, 58.3)			Stage		Total
			pTa/pT1	pT2	
Subtype	Basal	n	4	19	23
		%	17.4%	82.6%	100.0%
	Luminal	n	42	12	54
		%	77.8%	22.2%	100.0%
Total		n	46	31	77
		%	59.7%	40.3%	100.0%

**Table 3 cancers-15-00188-t003:** Association of BC molecular subtype with Grade.

*p*-Value = 0.025 OR = 4.24, 95% CI: (1.12, 16.06)			Grade (WHO 2004)		Total
			Low	High	
Subtype	Basal	n	3	20	23
		%	13.0%	87.0%	100.0%
	Luminal	n	21	33	54
		%	38.9%	61.1%	100.0%
Total		n	24	24	77
		%	31.2%	68.8%	100.0%

## Data Availability

The Cancer Genome Atlas (TCGA), Gene Expression Profiling Interactive Analysis (GEPIA).

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
