# Peer review of "Immunohistochemical Study of Bladder Cancer Molecular Subtypes and Their Association with PD-L1 Expression"

_cancers, 2022, doi:10.3390/cancers15010188_

Round 1

Reviewer 1 Report

I have read with great interest the paper by Goutas D. and colleagues. The manuscript addresses a very important topic of stratification of patients with bladder cancer to those who can potentially benefit by immunotherapy. As current means based on anti-PD1 tissue expression are suboptimal, the authors investigated PD-L1 related algorithms, as well as novel immunohistochemical biomarkers based on DNA mismatch repair proteins, in an effort to depict molecular heterogeneity (that is evident from the recent molecular clustering studies) but at the same time propose a protocol with increased clinical translatability/ applicability. Although the MMR expression results did not significantly contribute/ correlate with PD-L1, which decreases the novelty of the study in a way, still confirmation of the validity of PD-L1 related algorithms is an important finding.

I only have very minor comments to this well written study: 

1. I could not access the supplementary file, but a table with the list of the clinical and pathological figure for the recruited patients would be very informative. 

2. Were there any significant correlations of stainings with age and/or gender? 

3. You may consider to please use either comma or dot to indicate decimal places, in a consistent manner throughout the paper and figures. 

Reviewer 2 Report

In the present manuscript entitled “Immunohistochemical Study of Bladder Cancer Molecular Subtypes and their Association with PD-L1 Expression” the authors have assessed patients’ stratification into bladder cancer molecular subtypes, as well as the correlation of PD-L1 levels with disease clinicopathological features.

This is a significant study, however, there are minor issues that authors have to address to be able to publish their results.

Throughout the text:

1.       Please correct typos throughout the text. Τhe decimal point should be indicated with dot. Please replace the commas accordingly.

Introduction:

2.       Please reduce redundant information about cancer management in general. Focus on bladder cancer.  

3.       Please include the appropriate references in all periods.

4.       There is a more updated Global Cancer Statistics version (2020). Please update the reference 9.

Sung H, Ferlay J, Siegel RL, Laversanne M, Soerjomataram I, Jemal A, Bray F. Global Cancer Statistics 2020: GLOBOCAN Estimates of Incidence and Mortality Worldwide for 36 Cancers in 185 Countries. CA Cancer J Clin. 2021 May;71(3):209-249. doi: 10.3322/caac.21660

5.       Please provide more information on bladder cancer regarding 5-years prognosis of different subcategories, as well as current clinical need.

6.       Please provide more information about MMR proteins’ loss of function in bladder cancer onset and progression.

7.       Please provide more information about PD-L1 clinical value in bladder cancer management.

Materials and methods

8.       Please include a table with clinicopathological data of the screening cohort.

9.       Please include at least one Validation cohort. The authors recommend to download publicly available datasets and perform clinical evaluation, supporting their findings at transcriptional level, too.  

Results

10.   Please include a workflow diagram of the study.

11.   Please improve the Figures’ and Tables’ quality.

12.   Please include scale bars in Figures 2-5.

13.   Please clarify how you would classify patients with positive both GATA3(+) and CK5/6(+).

14.   The authors recommend to download publicly available datasets and perform survival analysis to strengthen their findings regarding PMS2 expression loss (e.g. TCGA, GEO dataset NCBI).

15.   How is the PMS2 loss related to PD-L1 status?  

Discussion

16.   The authors should discuss the limitations of their study.

17.   The authors should discuss more the translational impact of their findings for disease management based on more advanced literature.

Reviewer 3 Report

This is a well designed and well written study by Goutas et al. characterizing the molecular subtypes of urothelial carcinoma and their association with PD-L1 expression. The merit of this study is in its proposal of a simple pathologic algorithm to classify specimens as luminal or basal that is achievable in real-time, clinical arenas.

I have only a few comments:

1. The figure legends should describe the format of the data contained within the figure but not the findings themselves - that should be saved for the text. For example, the Figure 6 legend says "Figure 6. When evaluating PD-L1 expression based on TPS and IC score, a strong and consistent association with basal type bladder tumors was observed." 

2. Line 18-21 in the abstract: "Meanwhile, immune checkpoint inhibitors and their interference with the tumor-related immune-evasive strategies has led to the development of several immunotherapeutic drugs targeting programmed death-1 (PD-1) or programmed 20 death ligand-1 (PD-L1)." Immune checkpoint inhibitors haven't led to the development of immune checkpoint inhibitors - small point, but would just revise this sentence. 

3. Would include a section on limitations. At the end of the day, this is a small sample size and its generalizability is limited although it is hypothesis generating and it begs reproduction on a larger scale before really entering the realm of clinical relevance. 
